# The Preparation and Characterization of Chitosan-Based Hydrogels Cross-Linked by Glyoxal

**DOI:** 10.3390/ma14092449

**Published:** 2021-05-09

**Authors:** Beata Kaczmarek-Szczepańska, Olha Mazur, Marta Michalska-Sionkowska, Krzysztof Łukowicz, Anna Maria Osyczka

**Affiliations:** 1Department of Biomaterials and Cosmetics Chemistry, Faculty of Chemistry, Nicolaus Copernicus University in Toruń, Gagarin 7, 87-100 Toruń, Poland; 289185@stud.umk.pl; 2Department of Environmental Microbiology and Biotechnology, Faculty of Biology and Veterinary Science, Nicolaus Copernicus University in Toruń, Lwowska 1, 87-100 Toruń, Poland; mms@umk.pl; 3Department of Cell Biology and Imaging, Institute of Zoology and Biomedical Research, Faculty of Biology, Jagiellonian University, Gronostajowa 7, 30-387 Kraków, Poland; krzysztof.lukowicz@uj.edu.pl (K.Ł.); anna.osyczka@uj.edu.pl (A.M.O.)

**Keywords:** hydrogels, chitosan, glyoxal, tannic acid, cells study

## Abstract

In this study, hydrogels based on chitosan cross-linked by glyoxal have been investigated for potential medical applications. Hydrogels were loaded with tannic acid at different concentrations. The thermal stability and the polyphenol-releasing rate were determined. For a preliminary assessment of the clinical usefulness of the hydrogels, they were examined for blood compatibility and in the culture of human dental pulp cells (hDPC). The results showed that after immersion in a polyphenol solution, chitosan/glyoxal hydrogels remain nonhemolytic for erythrocytes, and we also did not observe the cytotoxic effect of hydrogels immersed in tannic acid (TA) solutions with different concentration. Tannic acid was successfully released from hydrogels, and its addition improved material thermal stability. Thus, the current findings open the possibility to consider such hydrogels in clinics.

## 1. Introduction

Currently, designing active substance delivery systems constitute one of the goals of novel materials development. The release of active compounds directly to the infected site provides the highest effectiveness of treatment [1]. Chitosan is one of the best-known polysaccharides that may be used to obtain biocompatible materials [2]. It may be isolated from food industry by-products as shells of crustaceans, which provides chitosan to be easily accessible. Essentially, material preparation methods that are cheap and fast should be considered since only those may find industrial applications.

Hydrogels based on chitosan have to be cross-linked to present improved material stability, as chitosan dissolves in aqueous-like conditions very easily. However, the solubility of chitosan depends on the molecular weight and deacetylation degree of chitosan as well as the pH of the solution. Various compounds are known to be applicable as cross-linker agents [3].

Materials may be enriched with active substances, for instance polyphenols. Tannic acid is the example of a polyphenol that shows numerous important properties. To our knowledge, the proposed tannic acid-loaded hydrogels preparation method is a novelty. Tannic acid provides antimutagenic, antitumor, and antioxidant effects [4]. Tannic acid has found industrial, pharmacological, and food applications. It is also used as an additive to medical products in burn treatment [5].

The term hydrogel is used to refer to the three-dimensional, cross-linked structures of hydrophilic polymers. Hydrogels are characterized by the ability to absorb large amounts of water or biological fluids. They balance the swelling process and form insoluble structures due to their three-dimensional network. Their affinity for water absorption is attributed to the presence of hydrophilic groups such as –OH, –CONH–, –CONH_2_–, and –SO_3_H in the polymers forming hydrogel structures. The water sorption capacity of the hydrogel depends on several factors, such as the nature and density of the polymer used and the degree of cross-linking. Hydrogel structures share some physical properties that resemble those of living tissues more than any other class of synthetic materials. These features are attributed to their high water content, their softness and consistency, and the low interfacial tension of water or biological fluids. Hydrogels may be used as wound dressings to promote wound healing, and they ensure the moisture environment with gas permeability. Such materials may be also applied to fill the bone defects providing they allow for the cells’ differentiation into osteoblasts.

Glyoxal is a chemical compound that contains two aldehyde groups. It is commonly used to cross-link polysaccharides [6,7,8]. It has cross-linking ability via acetal formation between the aldehyde groups of glyoxal and the hydroxyl groups of the glucosamine units of chitosan or through Schiff’s base formation between the free amino groups of chitosan and the aldehyde groups of glyoxal [9]. Our previous studies showed that chitosan/tannic acid mixture additionally cross-linked by glyoxal and lyophilized leads to the obtainment of porous biocompatible structures [10].

In this study, hydrogels loaded with tannic acid were examined for potential medical applications as wound dressings or compresses for burns. The hydrogel structure, tannic acid release as well as mechanical properties and thermal stability of hydrogels were studied. Moreover, the tannic acid release profile was studied. The biological studies included the cell culture test on hDPC cells by their viability measurement. The proposed method of hydrogel preparation is fast and cheap and does not require any equipment compared to method published by us previously [10].

## 2. Materials and Methods

### 2.1. Chitosan Characterization

Chitosan (low molecular weight) has been purchased from the Sigma-Aldrich company (Poznan, Poland). The deacetylation degree (DD, %) of chitosan and its viscosity-average molecular weight were determined by the using viscometer (ViscoClock equipped with Ubbelohde ( SI Analytics, Mainz, Germany). Determined DD was 78% [11], and the chitosan’s molecular weight be equal to 1.8 × 10^6^ D.

### 2.2. Sample’s Preparation

Glyoxal (G), acetic acid, and tannic acid (TA), commercially available compounds, were purchased from the Sigma-Aldrich company (Poznan, Poland). Glyoxal solution (40 wt % in water; M_w_ = 58.04 g/mol) was used. The molecular weight of tannic acid was 1701.2 g/mol. The chitosan solution at 2% concentration was prepared in 0.1 M acetic acid. 

Chitosan-based hydrogels were prepared by glyoxal addition to chitosan solution. Different chitosan/glyoxal ratios were tested. It was noticed that a high concentration results in prolonged the exceeding gelation time. The addition of glyoxal was 5 and 10%, where the most suitable glyoxal addition was 10%. Thereby, the glyoxal was added drop by drop in amount 10% based on the weight of chitosan solution (room temperature). After each cross-linker addition, the mixture was mixed slowly on a magnetic stirrer for 1 min and left without mixing for 15 min. As a result, semi-solid hydrogel in a gel-like form was obtained. Glyoxal is an aldehyde compound; it is important to not add more monomers that can react with chitosan, as glyoxal residues may be toxic. Thereby, hydrogels were then washed with distilled water to remove the unreacted cross-linker.

### 2.3. Scanning Electron Microscope

The morphology of the samples was studied using a scanning electron microscope (SEM) (LEO Electron Microscopy Ltd., Cambridge, England). Hydrogels were placed into plastic plates and frozen (−4 °C, 24 h). Then, samples were lyophilized (ALPHA 1–2 LDplus, CHRIST, −20 °C, 100 Pa, 24 h). Such prepared dry hydrogels were covered by gold (thickness 5–10 nm) and studied. Scanning electron microscope images were made with magnification 500× and 10.0 kV.

### 2.4. Thermal Properties

Thermogravimetric analysis and differential thermal analysis (TG-DTA) were performed at a heating rate of 10 °C/min from 20 up to 750 °C in the nitrogen atmosphere by using TA Instruments SDT 2960 Simultaneous TGA-DTA (TA Instruments, Eschborn, Germany). From thermogravimetric curves, the characteristic temperature at a maximum decomposition rate of the investigated composites was determined for each type of hydrogel.

### 2.5. Polyphenol Loading and Release

The prepared chitosan-based hydrogels cross-linked with glyoxal were washed by water and then immersed in a polyphenol solution at different concentration values: 2, 5, 10, and 20% (in water). The hydrogels were immersed for 24 h in 50 mL of TA solution (room temperature, sunlight exposure).

Hydrogels loaded with a polyphenol (tannic acid) were immersed in a phosphate buffered saline (PBS) solution (pH = 7.4). The content of the polyphenolic compounds was determined using the Folin–Ciocalteu test [12]. The collected samples (1 mL of each one) were mixed with 0.5 mL Folin–Ciocalteu reagent. After 3 min, 1 mL of saturated Na_2_CO_3_ solution was added. The prepared mixtures were incubated at 40 °C for 30 min. The absorbance was measured at 725 nm using a UV-Vis spectrophotometer (UV-1800, Shimadzu, Kyoto, Japan). The experimental calibration curve was prepared for the standard solution at the concentration 0–2.5 mg/mL (gallic acid; R^2^ = 09997). The samples were collected 15, 30, 45, 60, 120, and 180 min after immersion.

Several mathematical equations may be used to define the dissolution profile. Once an appropriate model is selected, the drug release profile can be correlated with active compounds release kinetic models e.g., zero-order model, model based on first-order kinetic equation, Higuchi model, or Ritger–Peppas model [13].

### 2.6. Mechanical Properties

Mechanical properties were measured by the use of a mechanical testing machine (Shimadzu EZ-Test EZ-SX, Shimadzu, Kyoto, Japan). Hydrogels were placed in a 24-hole plate to form cylindrical samples sized 20 mm in diameter and 13 mm in height for mechanical testing (Figure 1). The samples were introduced between two discs and compressed (the starting speed of 200 mm min^−1^; initial force of 0.1 N). The compressive modulus (Young modulus for the compression process), the maximum tension, and the percentage deformation at maximum tension were determined with the Trapezium X Texture program. The statistical analyses were made, and the results are shown as the average with the standard deviation.

### 2.7. Swelling Properties

Hydrogels based on chitosan cross-linked by glyoxal were prepared. Swelling properties were studies for hydrogels without tannic acid immersion as well as after immersion in 2, 5, 10, and 20% TA concentration for 24 h. Then, hydrogels were placed in the PBS solution (pH = 7.4). The weight of hydrogels was studied every hour until 8 h of immersion. Then, swelling ratios were calculated using the Equation:$$swelling\;\left[\%\right]=\frac{m_{t}-m_{0}}{m_{0}}\times 100\%$$
where *m_t_* is the weight of the hydrogels after the time of immersion in PBS and *m_0_* is the weight before immersion.

### 2.8. Blood Compatibility

A sample of anticoagulated (by CPDA-1) sheep blood (with 0.2 mL) was added to 10 mL of physiological saline solution containing different specimens at the same weight. Positive and negative samples were prepared by adding 0.2 mL of fresh blood to water and physiological saline, respectively. Hydrogels immersed without previous immersion in a polyphenol solution were used as control samples. All the test tubes were incubated at 37 °C. After 1 h, the suspension was centrifuged at 1000 rpm for 10 min [14,15,16]. The absorbance of supernatants was measured by a microplate reader, Multiscan FC (Thermo Fisher Scientific, Waltham, MA, USA), at 540 nm. Each sample was prepared in triplicate. The hemolysis rate was calculated using the Equation:$$rate\;of\;hemolysis\;\left[\%\right]=\;\frac{\left[OD\right]specimen\;-\left[OD\right]negative}{\left[OD\right]positive-\;\left[OD\right]negative}\;\times \;100\%$$
where *[OD]specimen* is the absorbance for samples, *[OD]negative* is absorbance for negative control (physiological saline), *[OD]positive* is absorbance for positive control (water). The experiment was carried out for hydrogels without washing, once washed by distilled water, without immersion in tannic acid solution, and after.

### 2.9. Dental Culture Studies

Dental pulp cells can be obtained from postnatal-, wisdom-, and/or deciduous teeth, providing a non-invasive alternative (compared to e.g., bone marrow) to obtain osteoprogenitor cells. Human dental pulp cells (hDPC) were obtained according to the method of Bakkar et al. [17] from the molar teeth of 31–43-year-old patients, both genders. Teeth were extracted due to orthodontic patients’ treatments and were considered as surgical waste. Studies were carried out in agreement with the Local Bioethics Committee at the Jagiellonian University in Kraków; approval No. 1072.6120.253.2017 and informed consent was obtained from all donors.

Before cell culture, a sterilization technique (i.e., exposure to 70% EtOH) was assessed, and hydrogels cross-linked by tannic acid were examined for the potential leaking of tannic acid to the culture medium. Then, the hydrogels were sterilized in 70% EtOH, followed by 10 min exposure to UV light and then left overnight to dry. Following sterilization, the hydrogels were placed in separate wells of 24-well culture plates and seeded with 2 × 10^4^ dental pulp cells (DPSC) suspended in 2 mL of culture medium. The latter was composed of alpha-minimum essential medium (αMEM) supplemented with 10% fetal bovine serum (FBS) and antibiotics (penicillin/streptomycin 1% mixture). The cells were assessed for viability after 24 h culture on hydrogels or on tissue culture plate (TCP). For cell viability assay (CellTiter96Aqueous One Solution Cell Proliferation Assay; Promega, Madison, WI, USA), hydrogels were washed gently and carefully with PBS, followed by the addition of 0.4 mL/well of 10% MTS reagent in phenol-free alpha-MEM. The plates were incubated at 37 °C until the apparent change of color from yellow to brownish. Then, the colored media were transferred to individual wells in 96-well plates and the absorbance was recorded at 492 nm using a plate reader (SpectraMax iD3, Molecular Devices, San Jose, CA, USA). The results were expressed as a percentage of live cells on the studied hydrogels vs. live cells on control hydrogel (100%) [18]. Data were analyzed by one-way analysis of variance (ANOVA) with Tukey’s post hoc test and *p* ≤  0.05 was considered significant.

## 3. Results

### 3.1. Scanning Electron Microscope

Scanning electron microscope allows observing the structure of hydrogels (Figure 2). Hydrogel without tannic acid has a regular porous structure with open interconnected pores. Immersion in tannic acid solution results in the change in hydrogel’s structure with closed pores.

### 3.2. Thermal Properties

The thermal behavior of hydrogels had been studied (Figure 3). The thermal properties of biopolymeric materials are interesting to be considered as biopolymers without cross-linking low denaturation temperature. Temperatures for maximum peaks were determined (Table 1). T_max_ (3) could not be measured for hydrogel immersed in 2% tannic acid solution, as the peak was not well visible on the TG and DTA curves. Three steps of degradation of hydrogel were noticed. The highest weight loss was determined in the temperature range 20–170 °C. It may be assigned to the elimination of water molecules present in hydrogel [19]. The shape of peak has been changed after the hydrogel immersion in tannic acid solution. As tannic acid has many hydroxyl groups, it interacts with water molecules, and their elimination is lower than from hydrogel without the presence of tannic acid. Two more regions in difference thermo gravimetric (DTG) curves may be distinguished. High peaks at 203.80 °C observed for hydrogel without TA decreased rapidly with the presence of tannic acid. This peak may be assigned to the degradation of polymeric structure (chitosan). In addition, peaks present in higher temperature (above 268 °C) are assigned to the chitosan degradation. It may be observed for each type of hydrogel (without and with TA). However, the maximum temperature has been shifted to a higher temperature after the loading with TA. It is a result of hydrogen bonds present between chitosan and tannic acid, which stabilize the hydrogel structure.

### 3.3. Polyphenol Loading and Release

After polyphenol loading (Figure 4), the hydrogel changed color as a result of tannic acid loading. The released tannic acid concentration was determined by spectrophotometric analysis. The cumulative percentage concentration of released tannic acid in square root time dependence is shown in Figure 5. It may be observed that the dependence of the released concentration in time is similar for each solution. The highest released concentration was noticed for hydrogels immersed in 20% solution. Initially, the difference was small, but in the end of the experiment, the released concentration value for 20% solution was three times higher than for 2% solution. The polyphenol release studies showed that solutions with higher active substance concentrations used for hydrogels loaded with tannic acid are more effective for cosmetic and medical applications.

Drug release data were fitted to the zero-order kinetic model and first-order kinetic model. In addition, the release data were fitted to the Higuchi and Ritger–Peppas equations to characterize the drug mechanism. The kinetic rate constants, release exponent n and values of R^2^ for each model were calculated by linear least-squares regression analysis (Table 2) [20]. The most suitable model was selected based on the highest correlation coefficient values. The R^2^ of the Higuchi model is higher than for other models. This implies that the kinetic of tannic acid release from chitosan/glyoxal hydrogel pursues the Higuchi square root model (Figure 5). For the Ritger–Peppas model, the estimation of n is utilized to describe the release mechanism. In our studies, the n is in the range 0.43–0.85. It suggests that the tannic acid release is controlled by both diffusion and relaxation of polymer chains [21].

### 3.4. Mechanical Properties

The mechanical properties of hydrogels were determined by the compression test (Figure 6). The stress–strain curves for hydrogels are shown in Figure 7. The Young modulus of hydrogel without immersion in TA had slightly higher values than after immersion. However, all observed changes were not statistically significant. Thereby, it may be assumed that the immersion in TA solution did not change the hydrogel mechanical resistance.

### 3.5. Swelling Properties

Hydrogels are materials with a high ability to swell. Hydrogels without immersion in TA solution showed high percentage swelling (Figure 8). Materials firstly immersed in TA solution had already swelled. Thereby, we did not observe a significant increase in the hydrogel weight. After 7 h of immersion in PBS, we observed a decrease in hydrogel weight, which may suggest that the dissolution process occurred.

### 3.6. Blood Compatibility

The hemolysis rate is one of the most critical criteria when considering materials in blood–material contact for biomedical applications (Figure 9). Materials that show hemolysis lower than 5% may be considered safe for biomedical applications [22]. The blood compatibility measurement results are listed in Table 3. Erythrocytes are sensitive to hemolysis due to shear stress [23]. The hemolysis rate for chitosan/glyoxal hydrogels without washing nor immersion in a polyphenol solution was 63.84%, which means that such hydrogel is highly hemolytic. However, after washing with distilled water, the hemolysis rate was negative. The hydrogels washing by water results in the removing of unreacted glyoxal, and the hemolysis rate decreased. A similar observation was noticed for hydrogels immersed in tannic acid solution. As hydrogels were initially immersed in distilled water, a nonbonded cross-linking agent was removed. Then, loading by tannic acid did not increase the hemolysis rate.

### 3.7. Cell Culture Studies

Our research has shown that the tested hydrogels immersed in TA solution do not show a cytotoxic effect on hDPC (Figure 10). A reduction in cell viability can be observed on hydrogels 2% and 20% in turn. The contents of 5% and 10% seem to stimulate the proliferation of cells grown on them. The hydrogel immersion in TA solution at concentrations of 2, 5, and 10% did not affect the increase of cytotoxicity of material.

## 4. Discussion

Hydrogels may be modified by the addition of different compounds to improve their physicochemical properties as well as biological response. Obtained hydrogels have a porous structure initially with open pores that after immersion in TA were closed. Chitosan-based hydrogels with gelatin showed better thermal stability than for pure chitosan. In addition, thermal properties were improved by TiO_2_ addition to the prepared hydrogels [24]. Tannic acid is able to form strong hydrogen interactions between polymer, both proteins and polysaccharides [25]. The two-step degradation of chitosan with TA hydrogels observed on TA and DTA curves may be assigned to the complex formation of chitosan-tannic acid (CTS–TA). The interactions between those two components have been considered in our previous work [26]. The complexation that occurs between chitosan and tannic acid result in the thermal stability improvement (Figure 11). The influence of chitosan complexation by alginate has been considered by Lv [27]. The authors strongly underline the importance of hydrogen bonds presence on the properties of the hydrogels as final products that are obtained.

The main application of hydrogels is as drug delivery devices [28]. We confirmed that a simple method of hydrogels immersion in tannic acid is effective for TA incorporation into a hydrogel. The immersion in TA did not affect the changes in the mechanical properties of hydrogels. Since chitosan-based hydrogels may be effectively soaked by tannic acid, they may deliver such active compounds as TA. Tannic acid may act as an anti-inflammatory factor, and it has antimicrobial and anticancer properties [29,30,31]. Chitosan-based materials have been considered as effective drug delivery systems of phenolic acids [32,33,34]. As a result, such materials enriched by polyphenolic acids may be used as food packaging or dressing materials as well.

As it was discussed by Liu [35], the introduction of phenolic units into the chitosan backbone is a great alternative to improve its physicochemical properties, but also, it remarkably increases material biological features. We reported that chitosan/tannic acid films show a selective influence on the cells that depends on their type—human bone marrow stromal cells (BMSC), human melanoma (MNT-1), human osteosarcoma (SaOS), and human keratinocytes (HaCaT) cells [36]. In the present study, we proposed chitosan-based hydrogels cross-linked by glyoxal. Glyoxal is an aldehyde that effectively cross-links amine groups present in the chitosan polymeric chain. Given that monomers of aldehydes are toxic, there is a need to determine the lowest amount of glyoxal to bind cross-linker completely to polymer [37]. It is important as unreacted cross-linker monomers may be toxic for the human body and should not remain in the final products and nonbonded.

There is information in the literature about the positive effect of TA-enriched scaffolds on osteogenic differentiation in BMSC cells [38]. Our research shows a stimulating effect on the proliferation of hPDL cells, which are also considered osteogenic [39,40]. However, checking the osteogenic properties of the studied hydrogels requires deeper biological analyses, which was not currently our goal in this article. Glyoxal was also studied by us before as a safe cross-linker useful to obtain stable scaffolds based on chitosan and tannic acid mixture [10]. However, it is necessary to remove underacted glyoxal as it may have a cytotoxic effect on cells.

## 5. Conclusions

Chitosan-based hydrogels were obtained successfully by the addition of glyoxal as a cross-linker. In addition, we have shown that polyphenols may be loaded to such hydrogels by simple hydrogel immersion in polyphenol solutions. Our studies suggest that obtained hydrogels may be used as phenolic acid delivery systems. All hydrogels were nonhemolytic and therefore they are safe to be applied in contact with blood. In addition, hydrogels did not show the cytotoxic effect. We believe the results of this work suggest that the obtained hydrogels may prove useful for several biomedical applications, such as wound dressings or bone-related treatment procedures.

## Figures and Tables

**Figure 1 materials-14-02449-f001:**
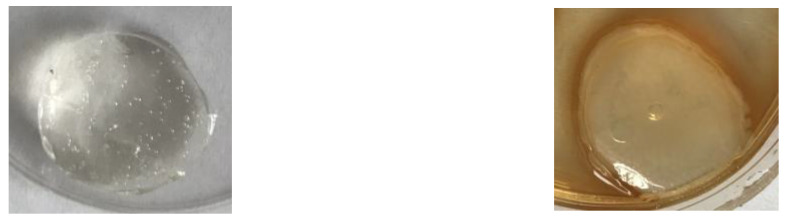
The shape of samples used for mechanical testing without immersion in TA (**left**) and after immersion in 20% TA solution (**right**).

**Figure 2 materials-14-02449-f002:**
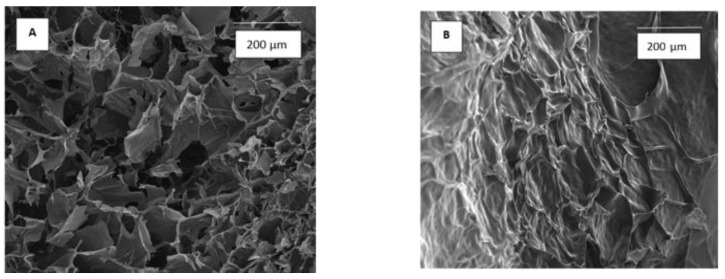
The structure of hydrogel without TA loading (**A**) and hydrogel loaded by immersion in tannic acid solution at 20% concentration (**B**); magnification 500×.

**Figure 3 materials-14-02449-f003:**
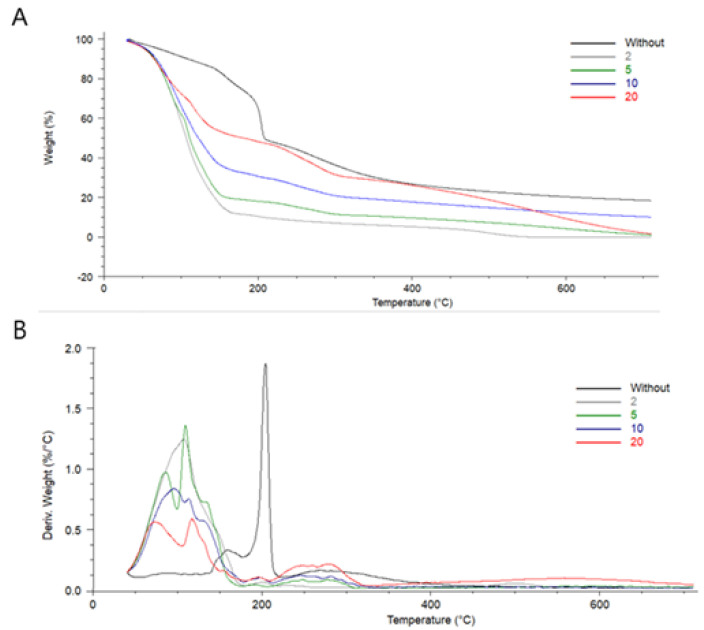
The TGA (**A**) and DTA (**B**) curves of chitosan-based hydrogels cross-linked by glyoxal without TA and after immersion in 2, 5, 10, and 20% tannic acid solution.

**Figure 4 materials-14-02449-f004:**
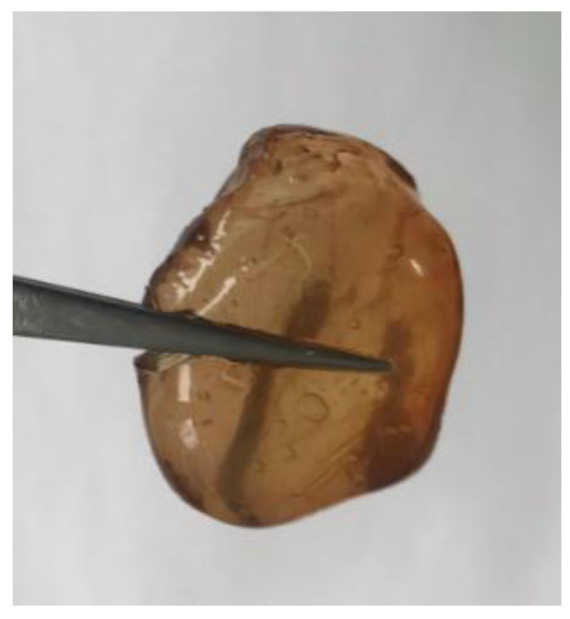
The hydrogel after immersion in 20% tannic acid solution.

**Figure 5 materials-14-02449-f005:**
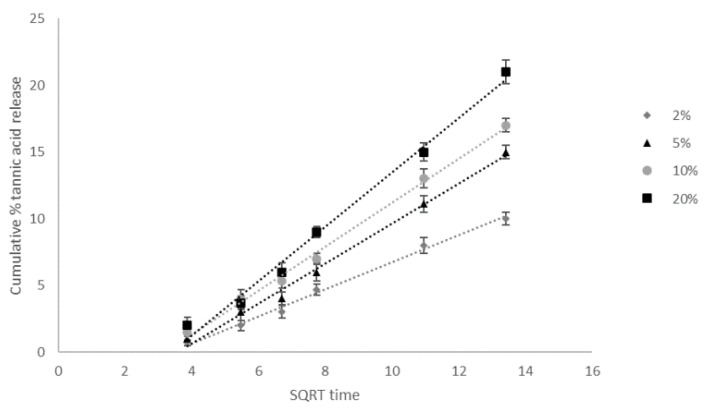
Tannic acid release from chitosan/glyoxal hydrogels curves of Higuchi square root model (n = 3).

**Figure 6 materials-14-02449-f006:**
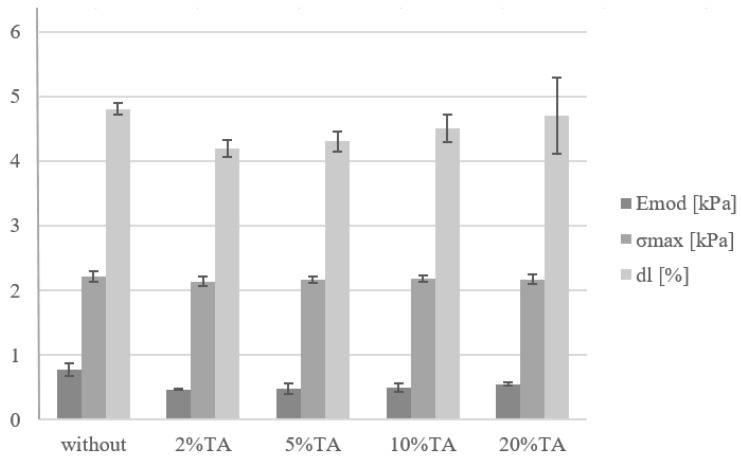
The mechanical parameters of hydrogel without immersion in tannic acid solution and of hydrogel immersed in TA solution at 2%, 5%, 10%, and 20% concentration; for *p* < 0.05, no statistically significant differences were observed (Emod—compressive modulus, σmax—maxium tension, dl—percentage deformation at maximum tension).

**Figure 7 materials-14-02449-f007:**
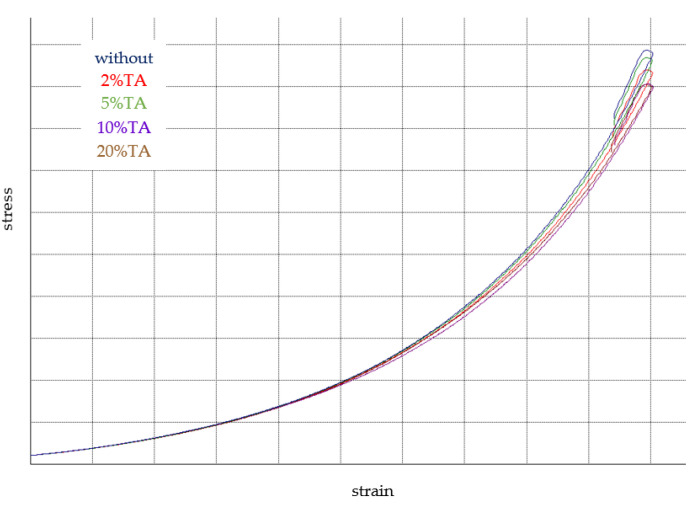
Stress–strain curves for hydrogel without immersion in tannic acid solution and of hydrogel immersed in TA solution at 2%, 5%, 10%, and 20% concentration.

**Figure 8 materials-14-02449-f008:**
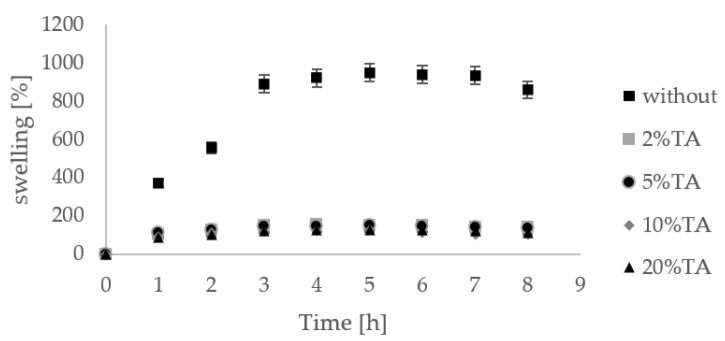
Swelling [%] of hydrogels (n = 3) without immersion in TA and after immersion in TA solution at 2%, 5%, 10%, and 20% concentration.

**Figure 9 materials-14-02449-f009:**
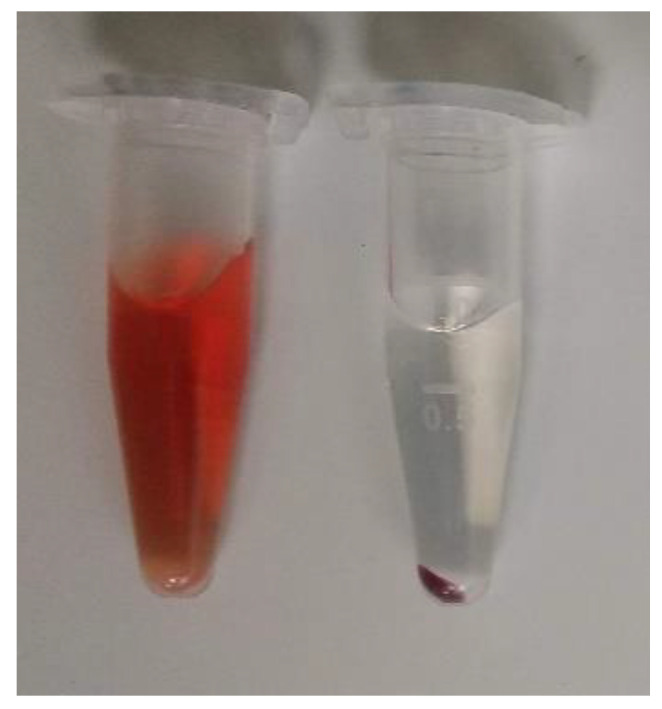
In vitro hemocompatibility assay left: hemolysis; right: no hemolysis.

**Figure 10 materials-14-02449-f010:**
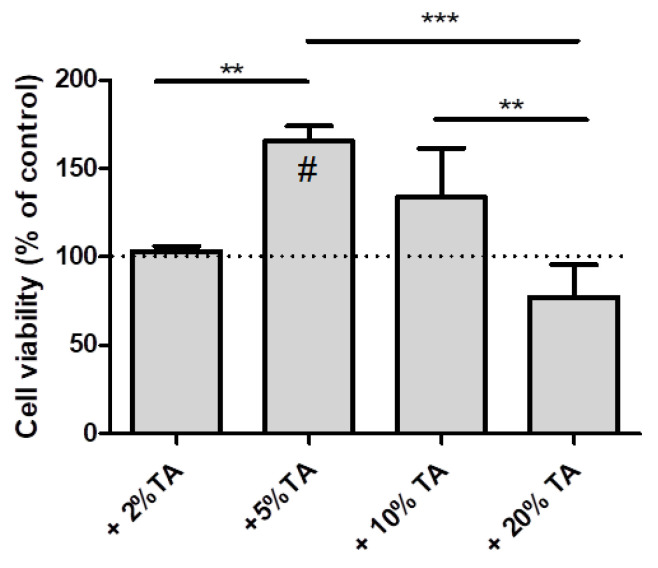
hPDL cells viability on tested hydrogels after 24 h of culture. Results are expressed as percentage change in cell viability compared to control hydrogel. # indicates significant increased from control. Statistically significant differences (** *p* ≤ 0.05; *** *p* ≤ 0.001) within the groups.

**Figure 11 materials-14-02449-f011:**
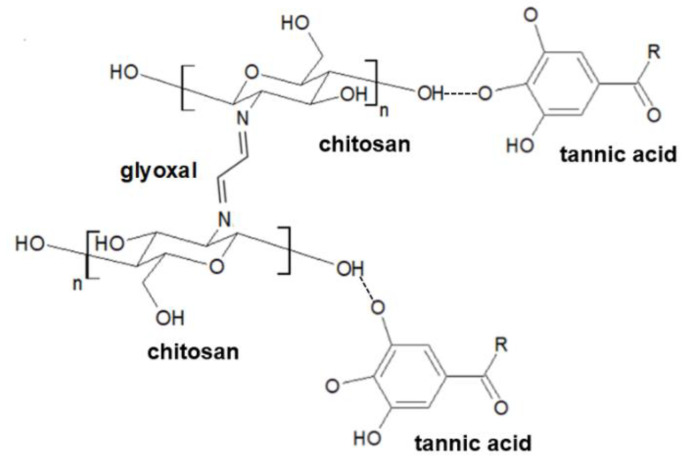
The schematic cross-linking mechanism of chitosan by glyoxal.

**Table 1 materials-14-02449-t001:** The results of DTA analysis with temperatures of maximum peaks

Specimen	T_max_ (1) [°C]	T_max_ (2) [°C]	T_max_ (3) [°C]
Without	159.33	203.80	268.95
2	107.21	200.52	- *
5	83.53	109.21	277.20
10	94.30	198.58	281.88
20	117.02	195.90	278.84

* The maximum temperature could not be measured.

**Table 2 materials-14-02449-t002:** Kinetic parameters of tannic acid release from chitosan/glyoxal hydrogels.

Polyphenol Solution [%]	Zero-Order		First-Order		Higuchi		Ritger–Peppas		
	k_0_	R^2^	k_0_	R^2^	k_0_	R^2^	k_0_	R^2^	n
2	0.0109	0.9213	0.0261	0.9331	0.1514	0.9942	0.6502	0.9611	0.6890
5	0.0118	0.9256	0.0269	0.9381	0.1531	0.9934	0.6513	0.9625	0.7011
10	0.0122	0.9209	0.0275	0.9405	0.1533	0.9957	0.6530	0.9604	0.7288
20	0.0125	0.9241	0.0281	0.9497	0.1542	0.9908	0.6538	0.9617	0.7301

**Table 3 materials-14-02449-t003:** Hemolysis rate for chitosan/glyoxal hydrogels after immersion in polyphenol solutions at 2%, 5%, 10%, and 20% concentration and without immersion for hydrogels washed and unwashed by distilled water.

Polyphenol Solution [%]	Hemolysis Rate [%]
Without	63.84 ± 3.80
Without/washed	−0.10 ± 0.09
2	−0.09 ± 0.07
5	−0.12 ± 0.11
10	−0.16 ± 0.08
20	−0.21 ± 0.15

## Data Availability

The data presented in this study are available on request from the corresponding author. The data are not publicly available due to 01.2023.

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
