# Peer review of "The Preparation and Characterization of Chitosan-Based Hydrogels Cross-Linked by Glyoxal"

_materials, 2021, doi:10.3390/ma14092449_

Round 1

Reviewer 1 Report

This manuscript lacks clear aim and experimental controls. The experimental choice were not relevant to a specific application. I have the following major comments:

1) Materials and methods: In sample preparation, why is it specified with sunlight exposure in hydrogel preparation? What is the role of sunlight?

2) 2.3. Scanning electron microscope: Details are missing. ´Covered with gold´- does it mean sputter coating with gold due to its high conductivity to achieve high quality images? It should be specified and also the thickness of the coating. Authors should also specify in what kV images were taken?

3) Blood compatibility study: Authors claim that this study was performed in relevance to wound dressing application. In that case, then its also important to study the biocompatibility with skin epithelial cells. One control experiment is missing here. Did authors tried to wash the hydrogels with buffer (Tris-HCL/Ammonium chloride or milliQ water several times) to remove unreacted glyoxal for the homolysis study? Because in the results authors claim that high hemolysis rate in chitosan cross-linked by glyoxal hydrogels is due to the unreacted glyoxal and this is not the case in polyphenol loaded hydrogels. A solid confirmation is missing in this study.

4) Figure 4: The study with Phenol release concentration. Did the authors performed this experiment in triplicates? Standard deviations are missing. Was the polyphenol loaded concentration calculated? Will the release increases with time? What happens after 200 mins will there be a saturation in the release?

5) Figure 6: This is contradicting the above result from hemolysis. In the above study, authors mentioned that hemolysis rate is higher due to toxicity from unreacted glyoxal in chitosan cross-linked by glyoxal hydrogels. Whereas, when it comes to ALP study, authors demonstrate that hydrogels indicate cell and no cytotoxic effect. What is the reason for it? Can authors explain why there is no effect from unreacted glyoxal in this case?

6) SEM of cells in hydrogels and TCP can be shown.

7) Thermal stability of Hydrogels: Authors should explain how this experiment is relevant for wound healing or bone related treatments? or specify why this characterisation is required for this study?

Author Response

Dear Editor,

On behalf of co-authors, I am submitting the revised manuscript materials-1188808 entitled “The preparation and characterization of chitosan-based hydrogels cross-linked by glyoxal” to be considered and subsequent acceptance for publication as “Original Article” in the Materials journal.

A revised version of the manuscript has been attached. Simultaneously, we would like to note that apart from reviewer’s valuable remarks, the authors placed additional editorial corrections, including references to improve the quality of the manuscript. All changes have been marked in red color as well as highlighted by “Track Changes”.

Herein, we greatly appreciate all the reviewer’s comments and we have thoroughly addressed to all of them point-by-point below:

Reviewer 1

1) Materials and methods: In sample preparation, why is it specified with sunlight exposure in hydrogel preparation? What is the role of sunlight?

Thank you for the comment. Sunlight does not play any role in the hydrogel preparation. We removed this aspect.

2) 2.3. Scanning electron microscope: Details are missing. ´Covered with gold´- does it mean sputter coating with gold due to its high conductivity to achieve high quality images? It should be specified and also the thickness of the coating. Authors should also specify in what kV images were taken?

Thank you very much for your valuable comment. We added details.

3) Blood compatibility study: Authors claim that this study was performed in relevance to wound dressing application. In that case, then its also important to study the biocompatibility with skin epithelial cells. One control experiment is missing here. Did authors tried to wash the hydrogels with buffer (Tris-HCL/Ammonium chloride or milliQ water several times) to remove unreacted glyoxal for the homolysis study? Because in the results authors claim that high hemolysis rate in chitosan cross-linked by glyoxal hydrogels is due to the unreacted glyoxal and this is not the case in polyphenol loaded hydrogels. A solid confirmation is missing in this study.

Thank you very much for the comment. We did not write it clear. Hydrogels were washed by water before immersion in TA. It is now corrected.

4) Figure 4: The study with Phenol release concentration. Did the authors performed this experiment in triplicates? Standard deviations are missing. Was the polyphenol loaded concentration calculated? Will the release increases with time? What happens after 200 mins will there be a saturation in the release?

Thank you very much for the comment. The SD is now added. After the time showed in Fig. 4 the hydrogel dissolved.

5) Figure 6: This is contradicting the above result from hemolysis. In the above study, authors mentioned that hemolysis rate is higher due to toxicity from unreacted glyoxal in chitosan cross-linked by glyoxal hydrogels. Whereas, when it comes to ALP study, authors demonstrate that hydrogels indicate cell and no cytotoxic effect. What is the reason for it? Can authors explain why there is no effect from unreacted glyoxal in this case?

Thank you very much for your valuable comment. Hydrogels were washed before the immersion in tannic acid solution. Thereby, unreacted glyoxal was removed. We apologize as we did not write it clearly in our previous version. The results showed that for such materials, the hemolysis was negative and the material may be classified as nonhemolytic. It is now corrected in the paper.

6) SEM of cells in hydrogels and TCP can be shown.

Thank you very much for the comment. We agree that SEM observation of cells on the hydrogels would be interesting. However, it is impossible in our lab as we may observe by SEM only dry surfaces. The drying process of hydrogels would damage its structure, so it would not be in our opinion, relevant to the type of material. Also, in our great knowledge, to compare to the biocompatibility of material before and after modification more relevant is to show results compared to the control which is unmodified material. We want to show differences in cell viability in contact with different types of material (before immersion in TA solution and after as dependence of the concentration  of TA solution). Results were shown in such form in many papers for example Łukowicz, Krzysztof, et al. (Materials Science and Engineering: C 109, 2020, 110535). We hope it is acceptable.

7) Thermal stability of Hydrogels: Authors should explain how this experiment is relevant for wound healing or bone related treatments? or specify why this characterisation is required for this study?

Thank you for the comment. Wound dressing with thermal stability is highly desired for burns healing (Issains et al. AIP Conference Proceedings, 2193, 2019, 020013). As biopolymers have low denaturation temperature, many researchers are focused on increasing the thermal stability of natural polymers-based films. In our opinion, the consideration of thermal properties of a material based on chitosan as natural polymers is an interesting aspect to study the influence of glyoxal on the thermal stability of the material. We discussed it in the paper.

Reviewer 2 Report

Authors have prepared chitosan-based hydrogel by using glyoxal and tannic acid for clinical applications. After careful evaluation, I have found that there is no much novelty in this work. Also, already authors have published one article based on this study (Materials Letters, 2021, 129667). However, this study may have potential for publications if authors consider these following major points for inclusion.

  1. Introduction section is very weak. Authors did not provided a specific challenge and motive related to this objective study. Also, why chitosan is more appropriate than other polysaccharides or biomaterials? Previous studies based on chitosan, tannic acid, and glyoxal? For example, Acta biomaterialia, 7(6), 2011, 2410-2417; Procedia Engineering, 110, 2015, 143-150; Materials Letters, 2021, 129667; etc. Here, authors should discuss previous published articles comparatively and why this study has more potential than those studies?
  2. In Fig. 2, TGA curves should also be included and discussed accordingly.
  3. 5 is little confusive. Please provide typical stress vs strain curves for compressive mechanical properties. Also, please provide digital images of the hydrogel samples used for mechanical testing for clear understanding of the product.
  4. For blood compatibility analysis, please provide digital image of this experiment to support this behavior and data provided in Table 2.
  5. For cell culture studies, authors should provide qualitative data in addition to quantitative data for effective evaluation. Also, MTT assay or Live/Dead assay should be performed and added to this analysis for confident conclusion.

After this revision, this study may be considered for publication.

Author Response

Dear Editor,

On behalf of co-authors, I am submitting the revised manuscript materials-1188808 entitled “The preparation and characterization of chitosan-based hydrogels cross-linked by glyoxal” to be considered and subsequent acceptance for publication as “Original Article” in the Materials journal.

A revised version of the manuscript has been attached. Simultaneously, we would like to note that apart from reviewer’s valuable remarks, the authors placed additional editorial corrections, including references to improve the quality of the manuscript. All changes have been marked in red color as well as highlighted by “Track Changes”.

Herein, we greatly appreciate all the reviewer’s comments and we have thoroughly addressed to all of them point-by-point below:

Reviewer 2

  1. Authors have prepared chitosan-based hydrogel by using glyoxal and tannic acid for clinical applications. After careful evaluation, I have found that there is no much novelty in this work. Also, already authors have published one article based on this study (Materials Letters, 2021, 129667). However, this study may have potential for publications if authors consider these following major points for inclusion.

Thank you very much for the comment. In the paper published in Materials Letters and in the paper submitted to Materials journal we used glyoxal. However, the materials are different and where obtained in different way. Both consider glyoxal as a cross-linker, however, we did not duplicate results or methods of sample preparation. We added this paper as a reference.

  1. Introduction section is very weak. Authors did not provided a specific challenge and motive related to this objective study. Also, why chitosan is more appropriate than other polysaccharides or biomaterials? Previous studies based on chitosan, tannic acid, and glyoxal? For example, Acta biomaterialia, 7(6), 2011, 2410-2417; Procedia Engineering, 110, 2015, 143-150; Materials Letters, 2021, 129667; etc. Here, authors should discuss previous published articles comparatively and why this study has more potential than those studies?

Thank you very much for the comment. In our opinion chitosan is the most widely used in biomaterials because it can be isolated from food industry by-products as shells of crustaceans what provides chitosan to be easily accessible. It is now written in the paper. Moreover, we discussed previous studies related to the chitosan/glyoxal materials.

  1. In Fig. 2, TGA curves should also be included and discussed accordingly.

Thank you very much for your comment. We added TGA curves.

  1. 5 is little confusive. Please provide typical stress vs strain curves for compressive mechanical properties. Also, please provide digital images of the hydrogel samples used for mechanical testing for clear understanding of the product.

Hydrogels were placed in a 24-holes plate to form cylindrical samples sized 20 mm in diameter and 13 mm in height for mechanical testing. The digital image may also be found in the paper. We decided to write the values of mechanical parameters as it allows to observe differences in all three parameters. Mechanical parameters were presented in such for before (e.g. Kaczmarek et al., J. Mech. Beh. Biomed. Mater. 110 (2020) 103916; Kaczmarek et al., Materials 13 (2020) 3419). Unfortunately, our testing machine is in the service (with the computer to operate). During the study we exported values not curve and thereby it is very difficult for us to put curves. We hope it will not affect the reviewer’s opinion about our paper. We appreciate your understanding.

  1. For blood compatibility analysis, please provide digital image of this experiment to support this behavior and data provided in Table 2.

Thank you very much for the comment. We added digital images of experiment. We hope it is acceptable now.

For cell culture studies, authors should provide qualitative data in addition to quantitative data for effective evaluation. Also, MTT assay or Live/Dead assay should be performed and added to this analysis for confident conclusion.

Thank you very much for your comment. In order to measure the viability of hDPC cells on the tested hydrogels, we used the MTS test (materials and methods section). It is an improved and faster version of the MTT test that uses color reactions. The injected cell reagent MTS (yellow tetrazolium salt) is metabolized by the mitochondria of living cells to give the brown formazan product. Cell viability is directly proportional to the intensity of the color change, which is then measured spectrophotometrically.

Reviewer 3 Report

The manuscript : The preparation and characterization of chitosan-based hydrogels cross-linked by glyoxal , is very general and i do not see any novelty. In addition the whole manuscript is very poor and need strong improvement before to be considered for publication. The methodologies, together with the results are poorly described, in addition the final aim of the study should be reported. Crosslinking chitosan with glyoxal to release tannic acid is an old story.

Some comments related to the manuscript: 

1)In the introduction part the authors

-report : chitosan have to be cross-linked to present the improved material
stability as chitosan dissolves in aqueous-like conditions very easily. This is quite general because the solubility of chitosan in aqueous solution depends on : pH, the Mw  and deacetylation degree.

- mention that hydrogels loaded with tannic acid were examined for potential medical applications. Could you report some example of possible suitbale application? 

-what ALP stands for?

2) section 2.1 Chitosan characterization : more details related to the viscometer used and the experimental condition have to be reported. In addition the molecular weight should be determined by GPC.  How the deacetylation degree has been evaluated by viscometer?

3) section 2.2 It is very confusing, please rewrite. More specification about preparation have to be reported (e.g volumes etc..)

4) section 2.5 : why tannic acid has been loaded after the hydrogel preparation and not during the preparation? 

5) section 2.8 ; the authors mention that the hydrogel was sterilized using 70% ethanol ; how ? it was immersed in ethanol? tannic acid is soluble in ethanol ( cc 100g/L) have you checked if some loss of tannic acid occurred? The application of UV could cause crosslinking between chitosan and tannic acid, have you evaluated it? 

6) section 3.3 : the amount of tannic acid loaded must be specified and the release trend analized using mathematical models to evaluate the driving force. In Figure 4 the best fitting should be reported with the relative equation and the kinetics parameters

7) the swelling properties of the hydrogel has to be determined. 

8) Figure 5. without; do you mean control? Add the y-axe in the figure

9) section 3.5 : the authors report: "The hemolysis rate for
chitosan/glyoxal hydrogels without immersion in a polyphenol solution was 63.84%. It suggests that hydrogel is highly hemolytic. It may suggest that glyoxal residues that did not react with chitosan are toxic However, after immersion in tannic acid solution, the material unreacted glyoxal is
removed and hydrogels have nonhemolytic properties"  . Immersion in tannic acid solution is not a justification of reduction in hemolysis, maybe even a simple washing to remove unreacted glyoxal could provide the same results. Please provide better explanation. 

10) It would be appreciate to report image of hemolysis studies 

11) stability of the hydrogel in particular possible lack of tannic acid has to be evaluated. 

Author Response

Dear Editor,

On behalf of co-authors, I am submitting the revised manuscript materials-1188808 entitled “The preparation and characterization of chitosan-based hydrogels cross-linked by glyoxal” to be considered and subsequent acceptance for publication as “Original Article” in the Materials journal.

A revised version of the manuscript has been attached. Simultaneously, we would like to note that apart from reviewer’s valuable remarks, the authors placed additional editorial corrections, including references to improve the quality of the manuscript. All changes have been marked in red color as well as highlighted by “Track Changes”.

Herein, we greatly appreciate all the reviewer’s comments and we have thoroughly addressed to all of them point-by-point below:

Reviewer 3

1)In the introduction part the authors

-report : chitosan have to be cross-linked to present the improved material

stability as chitosan dissolves in aqueous-like conditions very easily. This is quite general because the solubility of chitosan in aqueous solution depends on : pH, the Mw  and deacetylation degree.

Thank you very much for the comment. We absolutely agree. It is now commented in the introduction part.

- mention that hydrogels loaded with tannic acid were examined for potential medical applications. Could you report some example of possible suitbale application?

We propose hydrogels as wound dressings or compresses for burns as the tannic acid had been already studied in products for burn treatment (e.g. Khan et al., Chem. Biol. Interact. 125 (2000) 177-189).

-what ALP stands for?

In this version of paper (based on the reviews) we decided to removed ALP studies and replace it by more relevant experiment of MTS.

2) section 2.1 Chitosan characterization : more details related to the viscometer used and the experimental condition have to be reported. In addition the molecular weight should be determined by GPC.  How the deacetylation degree has been evaluated by viscometer?

Thank you very much for the comment. We added details related to the viscometer. The deacetylation degree was calculated by following the procedure (Motta de Moura, et al., Chem. Eng. Process 50 (2011) 351-355). We agree that the molecular weight may also be determined by different methods e.g. GPC. However, we are not able to carry out such tests in our lab. We believe that our method is acceptable.

3) section 2.2 It is very confusing, please rewrite. More specification about preparation have to be reported (e.g volumes etc..)

Thank you very much for the comment. More specific details are now written in 2.2.

4) section 2.5 : why tannic acid has been loaded after the hydrogel preparation and not during the preparation?

Thank you for the comment. The first aim of the study was to obtain chitosan/glyoxal hydrogel. Then study its application in soaking active compounds (in this paper tannic acid). Chitosan and tannic acid interacts by strong hydrogen bonds formation (Kaczmarek et al., Materials 13 (2020) 3641). Thereby, we decided to add glyoxal to chitosan without tannic acid to carry out the cross-linking process in the most effective way.

5) section 2.8 ; the authors mention that the hydrogel was sterilized using 70% ethanol ; how ? it was immersed in ethanol? tannic acid is soluble in ethanol ( cc 100g/L) have you checked if some loss of tannic acid occurred? The application of UV could cause crosslinking between chitosan and tannic acid, have you evaluated it?

Thank you very much for the comment. The sterilization method was checked by us before the experiment. We did not detect the presence of tannic acid in ethanol after immersion as well as did not observe any changes in material properties after exposure to UV light. Thereby, we may assume that the proposed sterilization method is safe.

6) section 3.3 : the amount of tannic acid loaded must be specified and the release trend analized using mathematical models to evaluate the driving force. In Figure 4 the best fitting should be reported with the relative equation and the kinetics parameters

Thank you very much for the comment. We calculated the cumulative percentage of tannic acid released from the material. The results are presented versus SQRT time as we have got the best fitting to Higuchi square root model. However, in Table 2 we presented the correlation coefficients also for zero-order, first-order and Ritger–Peppas model. We hope it is acceptable in the present form.

7) the swelling properties of the hydrogel has to be determined.

Thank you very much for the comment. We added this analysis to the manuscript.

8) Figure 5. without; do you mean control? Add the y-axe in the figure

Yes, without we mean hydrogel without tannic acid. It is now written in the figure description.  The Y-axe is now added. We did not write the units in Y-axis as they are different for Emod and Fmax, and for dl. We appreciate your comment.

9) section 3.5 : the authors report: "The hemolysis rate for

chitosan/glyoxal hydrogels without immersion in a polyphenol solution was 63.84%. It suggests that hydrogel is highly hemolytic. It may suggest that glyoxal residues that did not react with chitosan are toxic However, after immersion in tannic acid solution, the material unreacted glyoxal is

removed and hydrogels have nonhemolytic properties"  . Immersion in tannic acid solution is not a justification of reduction in hemolysis, maybe even a simple washing to remove unreacted glyoxal could provide the same results. Please provide better explanation.

Thank you very much for the valuable comment. We agree that the immersion in tannic acid solution may be a simple method to remover unreacted glyoxal. The discussion is now corrected.

10) It would be appreciate to report image of hemolysis studies

Thank you very much. We added the digital image of experiment.

11) stability of the hydrogel in particular possible lack of tannic acid has to be evaluated.

Thank you very much for the comment. We added the study of the swelling behavior of hydrogels without immersion in TA and after. To our great knowledge, it is one of the methods to study the stability of hydrogels.

Round 2

Reviewer 1 Report

I agree with the authors changes to the manuscript. I accept for publication. 

Author Response

Thank you very much.

Reviewer 2 Report

In my opinion, this manuscript can not be accepted in its current from.  

These are very strange points from the authors, that

  1. Authors have used chitosan, because it has widely been used by other researchers in this area. Also, materials used in this study and published paper (i.e. Materials Letters) are same with characteristics. 
  2. Whenever the characterization of samples is performed, at that time researchers keep all raw data from the used machine/instruments. Here, it is very strange that authors fetched only selected data for their convenience. It is not a good practice of research. Moreover, authors should provide actual stress-strain curves to show compressive strength and elastic modulus values. In addition, digital images of both samples with or without TA should be shown.
  3. for the efficacy of biomaterials for biomedical applications, both qualitative (e.g., live/dead assay) and quantitative (e.g., MTT assay) analyses should be performed and to be added. Also, why ALP activity was removed from this study? Only MTS or MTT assay is not relevant experiment. This is an approximate experiment. 

Author Response

Dear Editor,

On behalf of co-authors, I am submitting the revised manuscript materials-1188808 entitled “The preparation and characterization of chitosan-based hydrogels cross-linked by glyoxal” to be considered and subsequent acceptance for publication as “Original Article” in the Materials journal.

A revised version of the manuscript has been attached. Simultaneously, we would like to note that apart from reviewer’s valuable remarks, the authors placed additional editorial corrections, including references to improve the quality of the manuscript. All changes have been marked in blue color.

Herein, we greatly appreciate all the reviewer’s comments and we have thoroughly addressed to all of them point-by-point below:

Reviewer 1

  1. Authors have used chitosan, because it has widely been used by other researchers in this area. Also, materials used in this study and published paper (i.e. Materials Letters) are same with characteristics.

We appreciate you comment. As it may be read in the paper “Our previous studies showed that chitosan/tannic acid mixture additionally cross-linked by glyoxal and lyophilized leads to the obtainment of porous biocompatible structures (Kaczmarek-Szczepańska, et al Mater. Lett. 2021, 292, 129667).” We prepared scaffolds based on chitosan/tannic acid mixture with glyoxal as cross-linkers. The preparation method of the present and previous studies is different. Here we propose the preparation of samples without the need of using the lyophilizator. Also, we did not use chitosan/tannic acid complex only pure chitosan, then cross-linked and then we proposed obtained hydrogels as materials for active substance soaking and release (here tannic acid). The material characteristics are similar as they are proposed for biomedical materials. However, materials used to studies are different. We added the statement “The proposed method of hydrogel preparation is fast and cheap and does not require any equipment compared to method published by us previously (Kaczmarek-Szczepańska, et al Mater. Lett. 2021, 292, 129667).

  1. Whenever the characterization of samples is performed, at that time researchers keep all raw data from the used machine/instruments. Here, it is very strange that authors fetched only selected data for their convenience. It is not a good practice of research. Moreover, authors should provide actual stress-strain curves to show compressive strength and elastic modulus values. In addition, digital images of both samples with or without TA should be shown.

Thank you very much for the comment. We agree that it is not good practice and our mistake. Courtesy of another group we repeated studies with the use of different testing machine to obtain both parameters and curve (it is shown for average results). Thereby, we changed the method part of mechanical properties. Also, we added digital images of samples with and without TA used for analysis.

  1. for the efficacy of biomaterials for biomedical applications, both qualitative (e.g., live/dead assay) and quantitative (e.g., MTT assay) analyses should be performed and to be added. Also, why ALP activity was removed from this study? Only MTS or MTT assay is not relevant experiment. This is an approximate experiment.

Thank you for this valuable comments. Indeed, we have removed ALP assay as this was assessed for control hydrogel only and for cells cultured on a typical TCP – a comparison done for a general assessment of cells performance on control hydrogel vs. tissue culture plastic. This is though robust comparison, as the seeding and culturing cells on a hydrogel and TCP differ substantially and further controls/experiments are needed. Plus, as ALP assay opens a separate biocompatibility aspect referring to materials osteoconductivity/ osteoinductivity, we have assumed it may be too preliminary to discuss it in this manuscript as our studies regarding the above continue and more data can be presented in a separate study focusing on the materials potential to maintain/induce osteogenesis. Regarding the efficacy of biomaterials for biomedical application, we are aware that we cannot complete the evaluation of any material on MTS assay. Several additional qualitative/quantitative and molecular studies are required as we have presented in several of our other studies, especially regarding osteogenic potential of bioactive materials (Osyczka AM et al). However, the MTS test offers fast, quantitative and quite important primary assessment of any biomaterial cytocompatibility. And we now present MTS assay results after 24-h contact of cells with all studied control and experimental materials. As this work focuses on materials preparation and their physicochemical properties, this verification is added only to show the materials can be used for further evaluation in chosen/application-related cell cultures and eventually for their assessment regarding their clinical applications. We do hope this explanation satisfies the reviewer. The appropriate comment/discussion regarding the above has also been added to the manuscript text.

Reviewer 3 Report

The authors answer to all the questions and the improvement of the manuscript is evident. Then, I would reconsider my evaluation and accept the manuscript in the present form

Author Response

Thank you very much.